# “One Size Doesn’t Fit All”: Nutrition Care Needs in Esophageal and Gastric Cancer Survivors—A Qualitative Study

**DOI:** 10.3390/nu17091567

**Published:** 2025-05-02

**Authors:** Fatemeh Sadeghi, Juliette Hussey, Suzanne L. Doyle

**Affiliations:** 1Discipline of Physiotherapy, School of Medicine, Trinity College Dublin, D02 PN40 Dublin, Ireland; jmhussey@tcd.ie; 2School of Biological, Health and Sports Science, Technological University Dublin, D07 XT95 Dublin, Ireland; suzanne.doyle@tudublin.ie

**Keywords:** upper gastrointestinal cancer, cancer survivorship, oncology nutrition, nutrition care needs, quality of life

## Abstract

**Background/Objectives**: Nutritional complications are highly prevalent in esophageal and gastric cancer survivors and can negatively impact their quality of life. Gaining insight into survivors’ experiences with nutrition care can inform the development of tailored nutrition care programs for this population. The present study investigated esophageal and gastric cancer survivors’ nutritional challenges and complications. It also explored survivors’ and their carers’ perceived unmet nutrition care needs. **Methods**: Esophageal and gastric cancer survivors and their caregivers were invited to participate in a semi-structured interview. Participants were asked about their nutritional challenges and experiences with existing dietetic services in meeting their nutrition care needs. Data were analyzed using Braun and Clarke’s six-stage approach to thematic analysis. **Results**: Twelve individual interviews were completed and analyzed, and three themes emerged: nutrition-related challenges and complications, experiences with dietetic services, and coping strategies. Persistent gastrointestinal symptoms were reported as the main nutritional challenges impacting survivors’ daily and social lives. While participants reported good access to dietetic services, they emphasized the need for additional support during early post-surgery stages and long-term survivorship. Key unmet needs included access to personalized dietary advice for symptom management and timely information on nutritional challenges and post-treatment complications. **Conclusions**: The present study underscores the need for increased dietitian support throughout the esophageal and gastric cancer journey to provide timely, personalized, and practical dietary information for survivors and their caregivers. These insights should be considered for developing tailored nutrition care programs for esophageal and gastric cancer survivors.

## 1. Introduction

Esophageal and gastric cancers are among the most common malignancies worldwide, ranking 11th and 5th, respectively, in global cancer incidence [1]. In Ireland, they were the 14th and 10th most commonly diagnosed invasive cancers between 2019 and 2021 [2]. Advances in diagnosis and treatment have led to increased survival rates, as 5-year net survival rates increased by 12% for esophageal cancer and 14% for gastric cancer when comparing Irish data from 1994–1998 to 2014–2018 [2]. Despite substantial improvement in survival rate, cancer survivors often do not return to their pre-diagnosis health and suffer from long-term treatment side effects such as malnutrition, impaired physical function, and psychological problems [3,4,5].

While surgery remains the cornerstone of treatment in esophageal and gastric cancer, postoperative physiological changes in the GI tract can impair GI function and lead to long-term persistent side effects [6]. Vagotomy and anatomical alterations following esophagectomy and gastrectomy impair gastric motility, pancreatic enzyme secretions, bile salt absorption, and GI hormone regulations [6,7,8]. These changes can lead to delayed gastric emptying, early satiety, dumping syndrome, reflux, and diarrhea [6,7,8]. Dysphagia due to anastomotic strictures further limits dietary intake [6,9,10]. The loss of parietal cells and intrinsic factors affect B12 absorption, and alterations in the GI tract disrupt iron and calcium uptake, contributing to micronutrient deficiencies [6,11]. Thus survivors often suffer from nutrition-impact symptoms (NISs) such as eating and swallowing difficulties, anorexia, reflux, nausea, vomiting, diarrhea, dietary restrictions, and malabsorption, which can negatively impact their quality of life [6,12,13,14,15,16,17,18]. Thus, ongoing care and long-term follow-up are required to address the side effects of cancer treatment on survivors’ quality of life [19].

Given the significant relationship between nutrition and cancer, nutrition care should be considered an essential part of cancer survivorship [20]. Existing guidelines recommend key strategies such as regular malnutrition screening, preoperative and postoperative nutrition support, and delivering nutrition care within a multidisciplinary team to provide optimal nutrition care [21]. However, current evidence-based guidelines lack clear recommendations for managing long-term nutritional challenges in cancer survivors [21].

Several studies have investigated healthcare professionals’ awareness of nutritional considerations in cancer, their insight into the role of nutrition in cancer care, and the practicality of implementing nutritional guidelines in clinical practices [22,23,24,25]. It has been found that despite recognition of the importance of nutrition, barriers such as limited resources, knowledge gaps, and insufficient collaboration between oncologists and clinical nutritionists hinder the effective integration of nutrition care into patient care [22,23,24,25,26].

As recommended by the Multinational Association of Supportive Care in Cancer (MASCC) and the American Society of Clinical Oncology (ASCO), survivorship care should be person-centered and tailored to the survivors’ and their caregivers’ preferences and needs [27,28]. Understanding survivors’ experience of nutrition care services and their preferences can guide the development of tailored nutrition resources and services to address cancer survivors’ needs post-treatment [29]. Caregivers play a crucial role in supporting patients with cancer but often face significant psychological distress and challenges in adapting to their new responsibilities [30]. Therefore, gaining insight into their experiences with nutrition care can inform the development of tailored support services that more effectively meet the needs of both survivors and their caregivers.

Previous research in Ireland showed that cancer survivors consider nutrition an important part of cancer care, empowering them to take an active role in their health care [22,31]. However, they were vulnerable to misinformation and expressed a strong interest in receiving nutrition advice and information from trusted sources [32,33,34]. Notably, the majority of participants in these studies were breast cancer survivors, who face different nutritional challenges compared to esophageal and gastric cancer survivors. Thus, the present study aimed to investigate the availability of nutrition care services in Ireland and whether existing resources meet the nutrition needs of cancer survivors and their carers to expand current knowledge and awareness of their unmet nutrition care needs.

## 2. Materials and Methods

### 2.1. Eligibility

Adults over 18 years who were diagnosed with esophageal or stomach cancer or their caregivers or family members were eligible to take part in the present study. Fluency in English was necessary to be eligible for the current study. Participants had to be Irish residents, with treatment completed in Ireland.

### 2.2. Recruitment

The present study was advertised online across virtual social networks such as Twitter, Facebook, and Instagram to reach and identify interested participants. Additionally, it was advertised through Irish cancer charity organizations (the Irish Cancer Society and the Oesophageal Cancer Fund) via their social media platforms. The present study was also advertised in the St James Hospital UGI (Upper Gastrointestinal) clinic newsletter. The newsletter was posted to 200 patients listed in the clinic’s database using a simple random sampling method using a random number generator (random.org, accessed on 30 March 2022) [35]. It should be noted that all patients on the mailing list had previously consented to being contacted for research purposes. To confirm that participants were alive, we cross-referenced the clinic’s mailing list with the death notices posted on RIP.ie, a death notice website in Ireland, before sending out the newsletters.

### 2.3. Data Collection

Individuals were invited to participate in a semi-structured interview and were given the option to choose between remote or in-person interviews. In addition to qualitative data, primary quantitative data regarding demographics, cancer diagnosis, and treatment were also collected at the start of the interview. All interviews were conducted by a research dietitian (FS). It should be noted that research dietitians had no clinical involvement in the participants’ care and no affiliation with the hospitals where participants were treated. Additionally, to further minimize potential role-related bias, the interviewer’s position was stated in the participant information leaflet as research assistant and PhD candidate. The interviewer introduced herself in a research rather than clinical capacity at the beginning of each interview. Participants were assured of the confidentiality of their responses. A neutral, non-directive interviewing style was employed, with participants explicitly encouraged to share critical perspectives and informed that there were no right or wrong answers. The interview guide was adapted from a similar study [36]. Survivors and carers were asked about nutrition-related challenges they faced during and post-cancer treatment and their coping strategies. They were also asked about the availability of nutrition care during cancer treatment and recovery, their experiences with nutrition care, and their recommendations for its improvement. Data collection continued until saturation was reached. Data saturation was defined as the point at which no new themes or significant information emerged from the interviews, participants’ responses became repetitive in relation to the research questions, and additional interviews were unlikely to yield further insights into the identified themes [37].

The interviews were audio recorded and transcribed verbatim by FS. All identifiable data (names of participants, family members, clinical sites, and medical team members) were removed from transcriptions prior to analysis.

### 2.4. Data Analysis

Descriptive statistics were used to summarize participants’ characteristics. The interview transcript was analyzed using Braun and Clarke’s 6-stage approach to thematic analysis [38]. NVivo 12 software (QSR International, Malvern East, Australia) was used to code the interview transcripts. The analysis started with familiarization with data by reading and rereading each transcript and noting initial impressions. Then, initial codes were generated by reviewing and analyzing all individual transcripts. All quotes and phrases that captured or expressed the same or similar meaning and concept were set under relevant codes. In the next step, codes were grouped into initial themes relating to research aims. Interview quotes, codes, and themes were independently reviewed and analyzed by two researchers (FS and SD). FS and SD met and reviewed the preliminary analysis and completed joint coding, which involved merging or removing codes to reach a consensus on core themes and subthemes relating to the research question.

## 3. Results

### 3.1. Participants

A flow diagram of recruitment is presented in Figure 1. Initially, 19 individuals expressed interest in taking part in interviews. However, seven participants did not complete the interviews due to the following reasons:Two individuals (one survivor and one caregiver) did not meet the inclusion criteria as the cancer treatment was completed outside Ireland, and they were not Irish residents.Five individuals did not progress to enrolment as they did not respond to the researchers’ phone calls and emails after an initial expression of interest.

Consequently, twelve interviews were completed and analyzed in the present study. The recruitment of caregivers proved difficult as four of the seven individuals who were excluded after expressing interest were caregivers. While one did not meet the inclusion criteria, the remaining three did not respond to follow-up communication. The final sample consisted of nine esophageal and gastric cancer survivors and three family members of esophageal cancer survivors. Participant characteristics are presented in Table 1. Participants received treatment and nutrition care from seven different hospitals located in the east (4), west (2), and south (1) of Ireland. The average length of interviews was 46 min, ranging from 19 to 98 min.

### 3.2. Identified Themes, Subthemes, and Codes

Following data analysis, three main themes were identified: (1) nutrition-related challenges and complications, (2) experiences with dietetic services, and (3) coping strategies. Details of the subthemes and code structures are presented in Table 2. Both survivors’ and caregivers’ insights contributed to the themes. However, as caregivers represented a smaller portion of the sample, their insights were included where relevant to complement survivors’ perspectives. Thus, caregivers’ contributions are presented with an exploratory lens.

#### 3.2.1. Nutrition-Related Challenges and Complications

Participants reported various nutrition-related challenges. These included complications associated with gastrointestinal (GI) symptoms and challenges impacting individuals’ dietary habits, normal routine, and social life. Participants reported ongoing and long-lasting nutrition-related and GI symptoms from the start of their cancer journey and diagnosis through to survivorship. Diarrhea, swallowing difficulty, and dumping syndrome were frequently reported symptoms.

Participants also highlighted challenges for compliance with oral nutritional supplements (ONSs). Taste fatigue, GI disturbance, the prolonged and repetitive consumption of ONSs, and a dislike of taste and texture were reported as factors that limit compliance with the ONS regimen: “I kind of eased off the (ONS drink name) because you kind of get sick of the same flavors.” [P2]; “She (dietitian) given me some protein drinks, and they weren’t really agreeing with me.” [P5]. “Supplements were very heavily tasted, although there were maybe fruit flavors, it was, it was still heavy creamy.” [P11].

Nutritional challenges and complications impacted survivors’ daily routine and social life post-treatment. Several participants highlighted the challenges associated with meal preparation and altered meal plans. Participants noted meal preparation as a time-consuming and effortful task due to altered dietary patterns post-treatment, which interfered with everyday life, work, and leisure time. Also, the significant burden on caregivers in relation to meal preparation was reported: “It’s just you feel as if you’re eating all the time or preparing food, oh that’s tough I have to say.” [P2]; “I find that my whole thinking every day revolves around, a lot of it, revolves around, you know, what am I going to eat next or what time will I eat.” [P4].

Participants also reported limited food choices post-surgery due to fear of postprandial complications or prioritizing low-volume and high-calorie food choices to prevent weight loss: “This is the same thing again is all. It looks so boring when you write it all down……. That’s just become so unusual to sit down to, you know, a salad. Normally, if I go out to my dinner and put salad on the side of the plate, somebody else will eat it because I’m not interested in using up that space for the salad.” [P2].

Nutritional challenges and complications impacted survivors’ social life post-treatment. Participants discussed their concerns about socializing and dining with others due to GI disturbance and their special dietary requirements. They also noted the unpredictability of nutrition-related symptoms and how this impacted their daily routine as well as planning for leisure activities and socializing: “You know the thing is, I don’t drink, so going out for a meal used to be our social thing. You know, its suddenly going out for a meal, it’s a bit of, a bit of an issue.” [P2]; “We didn’t eat out, and then we did, but very much just together, never, never back with friends.” [P7—caregiver]; “You’d be avoiding traveling like that if you couldn’t get to the bathroom fast enough.” [P3].

Some participants reported a lack of awareness and an unwillingness of hospitality staff to accommodate the dietary restrictions of cancer survivors, which limited dining out and socializing around food: “If you say to them, hey, I can’t eat a lot, so please just give me half a portion. Uhm, restaurants don’t like that. They don’t, they don’t take that, and they always assume that you just want to pay less. It it’s a challenge.” [P7—caregiver]; “If I asked for the children’s menu because it would be smaller, they wouldn’t give it to me because I’m an adult, like very few places would accommodate.” [P8]. Also, participants voiced the psychological burden associated with special dietary considerations and symptoms as a barrier to social eating: “You know, anxiety that something will happen could happen, that he could choke, that he might choke.” [P10—caregiver]; “It is embarrassing going out with other people, and they’re having a three-course meal and here’s me stuck with the main course.” [P9].

#### 3.2.2. Experiences with Dietetic Service

The majority of participants reported a high degree of accessibility to dietetic services, particularly after surgical resection. Providing dietetic care simultaneously with cancer surveillance visits and remote dietetic services facilitated access to nutrition care: “I used to e-mail them (dietitians and nurses); nothing was ever a panic. I’d e-mail them, and they get back in touch with me within a few days.” [P2]; “I have their (dietitians) number, and if I had any problems, I could always give a ring if I wanted to” [P3]; “They’re (clinical team including dietitians) always there if I ever had a worry or a query, an e-mail or a call, and they’re straight back to me.” [P4]; “Every time I visit the hospital to see the surgeon,… I would see the dietitian …. If I needed a dietitian, all I had to do was ring and make an appointment.” [P11].

The majority of participants reported positive feedback on their dietetic visits and found sessions helpful in answering their questions and concerns: “The dietitians, they were very good, they were very attentive.” [P2]; “We were talking through where he (survivor) was with food or what he was doing, where we were going, those kinds of things, those kinds of meetings were very useful for me.” [P7—caregiver]; “Every question was answered, and we were also given additional information to take home.” [P10—caregiver].

However, when asked about the adequacy of dietitian contacts, participants expressed a need for additional contact with dietitians during the early stages of post-surgery. They also stated a need for extended survivorship care and contact with dietitians to manage the long-term and late effects of surgical resection: “If it had been the case where they followed up, maybe more often, as I said for the month or two months afterwards, it probably would have helped me a little bit more.” [P4]; “It (dietitian visits) came to an end after a year, I felt this should continue for longer…I just feel like you’re on your own after a certain amount of time, you know.” [P9].

Despite positive feedback on the benefits of dietitian visits, participants discussed unmet information needs. Participants noted the necessity for practical, personalized, tailored dietary advice to improve symptom self-management: “It’s not like a one leaflet fits all or one-size-fits-all. If it could be kind of tailored to suit the type of surgery that the person had, you know, it might benefit individuals better in, in my own opinion.” [P4]; “Everybody’s different, you know, it’s different for everyone. One size doesn’t fit all……practical things that you can go out and buy in the shop like the protein yogurts and fortified milk.” [P5]; “They talk about having snacks and things like that, you know. Well, what exactly is a snack?” [P6]; “The advice was always very much the same. You know, just try, have high calorie-dense food, and you know that just wasn’t appealing.” [P7—caregiver].

In addition, participants reported that they received limited information on nutritional challenges and common complications post-treatment. This information was deemed essential to facilitate adopting dietary modifications post-treatment. Participants also reported that this information may provide insight for recognizing common versus major complications post-surgery, which might require clinicians’ input: “It would have been nice to have a little bit more support, a little bit more information about what to expect.” [P1]; “Receiving a little bit more clear information about the biology and how your body has changed and what’s the reason that you need to do this modification in your diet after the surgery…. you know what is normal and what isn’t.” [P5]; “I didn’t receive any information about swallowing or choking. This was, this was a new experience for me, I believe that if there were, you know, if it’s common that people choke after gastrectomy and the esophagostomy, then there should be some information there to indicate… at least give the patient an idea of what’s ahead of them.” [P11].

When asked for feedback on information received (i.e., written information, books, leaflets, etc.), participants suggested additional educational material and methods such as videos, trusted online resources, and online forums to allow live interactions between survivors and dietitians: “Maybe a few short little videos made that would back up the leaflets. …. if you could visually see the person with the stomach cramps, maybe run into the toilet, you know, different things like that, you kind of know what is ahead of you because sometimes reading it, it doesn’t have the same impact actually.” [P4]; “YouTube videos and how to, you know, on recipes, easy recipes to make yourself in the early days.” [P5]; “If there was a website or, you know, someplace I could log into that would have that information just all in one place. I tend to be a bit messy, and I pick things up and put them down and forget where I put them.” [P6].

Additionally, participants highlighted the benefits of continuous education and reiteration of dietary recommendations to improve compliance. It also was noted that complications at the acute phase post-surgery might limit the absorption and retention of dietary information. Therefore, repeated self-education and reminders from dietitians were suggested to improve adherence to dietary advice: “You know, I was very ill at that stage very, very ill, and I found it very hard to take everything that was coming at me on board….it was a lot for me to take on board, you know. So, I tended to, you know, I picked up some of it and then I would try and look it up on the internet.” [P6]; “You would read something and go, I really do need to implement these smaller meals and more often because you kind of keep falling back into this view of I’m OK again, so I’m normal again, so…. I think your brain goes back to what it used to be.” [P7—caregiver].

#### 3.2.3. Coping Strategies

When participants were asked about how they managed their GI symptoms and nutrition-related complications, several coping strategies to deal with post-treatment complications were described. These included practical and suggested strategies, peer support, and additional support for family members and caregivers.

Following dietary recommendations and modifying food choices were attempted to prevent and manage complications. Eating small, frequent meals, avoiding drinking at mealtimes, avoiding certain food items, and safe swallow strategies were often reported to reduce gastrointestinal symptoms.

Acceptance was highlighted as a key strategy for adapting to changes and new life post-treatment: “You just adapt. Just over time, you know, it’s just the new normal, and it’s OK.” [P5]; “We’ve made it the norm that that’s the way it is. So, at this stage, it’s just accepted, you know.” [P10]; “I don’t believe I’ll, I’ll ever be able to, to get back to where I used to be. I’ve come to terms with that.” [P11].

Availing of support from family and caregivers for meal preparation and purchasing ready meals was suggested to improve food access and nutritional adequacy post-treatment. It also may reduce the burden associated with meal preparation: “If you get somebody to cook for you, … would be great. It’s a big help…the pre-prepared meals, something like that, that might help somebody.” [P2].

Participants also discussed their strategies for limiting the impact of treatment on their social life by considering changes in their social dining habits. These included requesting smaller portions or one-course meals and adjusting the type, time, and duration of social dining: “If I go to a restaurant, I’ll ask for the children’s menu.” [P1]; “You might go out, and you just have a starter instead of a whole meal.” [P2]; “So, we know if we’re meeting friends, we just go for an early bird in a very kind of concise short window of time and it’s great.” [P7—caregiver]; “I would always order the main course. I would never have the starter or the dessert because I just cannot eat that amount of food.” [P9]; “We would look at the menu very carefully and choose very carefully.” [P10—caregiver].

When asked about insights or experiences of attending support groups, some participants described the benefits of joining support groups, such as peer learning and emotional support for coping with the side effects of treatment: “You could actually talk to others that were in the same boat, or you could discuss, you know, how you are getting on with your food intake or what works for you and then you know what works for someone else…. what people are able to eat or hints or tips that others might have.” [P4]; “It (support group) does help because, like, you know you’re not the only one.” [P8]. However, availability and access to support groups varied depending on geographical location and the type of cancer. Establishing virtual support groups was also suggested to improve access to peer support: “It would be good if there was a support group in Ireland for stomach cancer survivors…. I’d love to talk to somebody [who has] been through the same journey as me. So, like that would have really helped.” [P5]; “Probably WhatsApp group, which may be backed up with a possibility of an online meeting every now and again, or even a, you know, the group get together at some stage.” [P6];

Participants also highlighted the lack of information on available resources for caregivers and family members. Introducing available resources at an appropriate time was suggested to support caregivers: “I think to know that there were some more resources that were just there, you could have availed of them if you’d known. I think all of that just a little bit earlier on would make a huge difference for the, for the people who are part of the support network of the patient.” [P7—caregiver]; “You don’t realize until things go bad that you need support. In the good times, you go fine, don’t need, everything’s fine, grand you know, and things go bad, you go. Oh, my goodness. Where do I turn? Who do I turn to?” [P10—caregiver].

## 4. Discussion

This study is the first to explore the experiences of esophageal and gastric cancer survivors and their caregivers regarding dietetic services in Ireland, their post-treatment nutritional challenges, and their preferences for nutritional care. Four key findings emerged:Cancer survivors continue to experience long-term effects that significantly impact their daily and social lives.While dietetic services are available during the acute recovery phase, there is a clear need for more intensive support during this period and extended follow-up care.Several unmet nutrition care needs were identified, including limited information on post-treatment complications, a lack of practical and personalized dietary advice, a lack of innovative and interactive education tools, and the necessity for repeated nutrition education.Survivors suggested several strategies to manage the adverse effects of treatment, including dietary modifications, acceptance, peer support, and additional resources to assist both survivors and their caregivers.

Figure 2 illustrates the key findings of the present study, including challenges, coping strategies, unmet needs, and proposed solutions.

### 4.1. Persistent GI Disturbance

In the present study, participants reported persistent GI disturbance, which they attempted to manage with dietary and lifestyle modifications. This finding was in line with a previous study that reported the implementation of dietary modifications by cancer survivors to limit nutrition-impact symptoms (NISs) [39]. Participants also spoke of accepting and adopting a “new normal” as another coping strategy, which is a common theme in the literature exploring patients’ experiences after esophageal and gastric cancers [40,41,42]. Whilst some level of adaptation is to be expected, it must be considered that some of these symptoms do not need to be simply tolerated and may, in fact, persist as a result of inadequate dietetic support and follow-up. Aligned with previous studies, the results of the present study demonstrated that persistent and unpredictable NISs were associated with emotional distress and anxiety in social situations involving eating [43,44]. Long-lasting NISs beyond the acute phase of survivorship can impact cancer survivors’ physical, emotional, and social function [42,45]. Post-operative follow-up has traditionally focused on tumor recurrence and survival, but given the significant and long-lasting burden of GI symptoms following esophago-gastrectomy, there is an increasing need to monitor and address GI symptoms and complications to enhance survivors’ quality of life [46].

### 4.2. Accessibility and Effectiveness of Dietetic Services

The majority of participants in the present study acknowledged the helpfulness of dietitian input and reported that dietetic services were generally accessible. This finding is consistent with a national survey that reported positive views regarding the role of nutrition in cancer care, high ratings of the helpfulness of dietetic input in cancer survivors, and greater access to dietetic services in gastrointestinal cancers [31,47]. However, the need for repeated dietary education, extended dietetic reviews and follow-ups, frequent contact with dietitians in the acute phase post-discharge from the hospital, and practical and timely information was highlighted in the present study. Limited dietary follow-up led participants to seek dietary advice from other sources, which may be complicated by the difficulty in identifying reliable information sources [48]. Previous studies have suggested that frequent exposure to nutritional information can lead to dietary modifications and highlighted the need for repeated nutritional education to address cancer survivors’ dynamic needs [49,50]. Regular contact with a dietitian has also been shown to improve self-confidence in symptom management and lead to positive behavioral changes in gastrointestinal cancer survivors [51,52].

It was noteworthy that participants reported receiving limited information and guidance about common postoperative complications and symptoms. This aligns with findings from Bennett et al., where esophageal cancer survivors reported a lack of information about what to expect throughout cancer treatment and survivorship and highlighted the importance of this information to empower patients and their families to prepare for challenges during the cancer journey [53]. Preoperative nutritional consultation has been shown to positively impact unintentional weight loss and oral intake and reduce readmissions due to tube-feeding-related complications in esophageal cancer survivors [52,54]. Thus, providing information on treatment and its associated side effects should be considered in standard pre-operative care to prepare patients and their caregivers for managing potential symptoms and improving overall treatment adherence.

### 4.3. Resource Limitations and Alternative Approaches

As noted above, the findings of the present study highlight a significant need for intensive dietetic input with pre-operative education, frequent support during the acute post-operative recovery, and extended dietetic follow-up into long-term survivorship. However, resource limitations can impact the frequency and continuity of contact with dietitians [51,55]. This is particularly relevant in the current Irish context, where oncology dietetic services are severely under-resourced [47,56]. In 2019, the Irish Nutrition and Dietetic Institute (INDI) reported to the National Cancer Control Program (NCCP) that the dietitian-to-patient ratio was far below the desirable level of 1:120, with only one dietitian per 4500 cancer survivors in Ireland [47,56,57]. In light of resource limitations, dietetic service providers should consider various approaches to care delivery. Alternative approaches to improve care delivery, such as telephone consultations, virtual group education, and online forums and dedicated websites, were suggested by participants. Respondents also suggested access to online interactive information sources and visual education material to enhance the effectiveness of nutritional education. Interactive forums have been recommended as cost-effective strategies to deliver nutrition care amidst a stretched dietetic service and have previously been reported as the preferred method for receiving nutritional information and support by Irish cancer survivors [58,59].

### 4.4. Reviewing Standard Dietary Advice

The standard dietary advice provided to esophageal and gastric cancer survivors should also be reviewed in light of the findings of this study. For example, oral nutritional supplements (ONS) remain the cornerstone of dietary advice for poor appetite and malnutrition [60,61,62]. However, it was evident in this study that they were poorly tolerated and complied with, which will negate their benefit. Factors affecting adherence to ONS in patients with cancer include changes in taste and smell, swallowing difficulty, GI disturbance, palatability, and prolonged ONS consumption [63]. Several strategies are recommended to improve ONS compliance, including considering sensory changes in cancer in designing ONS products, optimizing ONS texture and flavor profiles, and providing ONS products based on individual preference [63,64]. However, these strategies can only be effective as part of a framework where there is regular dietetic contact with survivors and caregivers throughout treatment and survivorship to adapt a personalized care plan.

Participants also emphasized the need for personalized and practical dietary advice. Receiving generic dietary advice and a lack of practicality were reported in previous studies involving Irish and international cancer survivors [32,48,59,65]. Practical, individualized dietary guidance that considers the real-life challenges of meal preparation is essential for ensuring better nutritional outcomes for cancer survivors. In a study by Wang et al., difficulty in meal preparation was reported as a predictor of poor nutritional status in gastric cancer survivors [66]. In the current study, survivors and their caregivers highlighted the burden of meal preparation and the lack of practical dietary advice and information to help manage this challenge. To address this issue, participants in the current study suggested that caregiver assistance with meal preparation and access to ready-made meals could help reduce the burden. Similarly, previous studies suggested that formal and informal support, ready meals, and food delivery services can support meal preparation and prevent dietary inadequacy [36,67]. Therefore, it is recommended that post-treatment dietary advice be individualized, with particular attention given to meal preparation challenges to ensure effective nutritional care.

### 4.5. Peer Support

Access to support networks has an important role in coping with psychological challenges in cancer survivors [65]. In this study, participants highlighted limited access to peer support and emphasized the benefits of such support. In line with the participants’ views, emotional support and transferring knowledge and experience are suggested as benefits of peer support in cancer [44,68]. However, it is important to consider the potential challenges in developing support groups, including the negative impact of observing the deterioration of peers’ health, diverse information needs, group dynamics, leadership, and sustainability [44,68].

### 4.6. Support for Carers

Participants also stressed the importance of timely access to information about available services and support for caregivers. Caring for a loved one with esophageal cancer was reported to be stressful and restrictive and often can change caregivers’ daily routines [30]. Indeed, the adverse impact of cancer treatment has been shown to affect carers’ social life and their relationships [69]. Therefore, it is essential to consider caregivers’ needs and implement strategies to prepare caregivers for supporting cancer survivors and also address their needs [30,70]. Providing a list of available psychosocial support and other resources at the time of diagnosis is recommended to better support patients and their caregivers throughout the cancer journey [71]. A recent systematic review of studies on carers’ experience of supporting UGI cancer survivors indicated the benefits of social, psychological, and practical resources on mediating carers’ burden [72]. Providing caregivers with psychoeducational programs, skills training interventions, and therapeutic counseling has been shown to significantly reduce caregiver burden, enhance their coping abilities, boost their self-efficacy, and improve various aspects of their quality of life [73].

### 4.7. Strength and Limitations

This study provides valuable insights into nutrition care needs in esophageal and gastric cancer survivors. However, the present study has several strengths and limitations that should be considered when interpreting the findings. One of the strengths of this study is the diversity of participants who completed their treatment in seven different hospitals located in various regions in Ireland. This can reduce the bias associated with differences in care received and capture common unmet needs in esophageal and gastric cancer survivors in Ireland. Also, the majority of participants were >2 years post-treatment, which allowed the investigation of nutrition care needs in long-term survivorship. Furthermore, the semi-structured design of interviews and the completion of all interviews by a single researcher ensured consistency in data collection. Moreover, the interviews were completed by phone at the survivors’ convenience while in their own environment, perhaps creating a more comfortable situation for reflecting on their survivorship challenges [74]. Additionally, as all participants opted for telephone interviews, this may also reflect the perceived convenience, increased privacy, or greater sense of comfort with discussing personal health issues in a remote setting [75]. Also, collecting qualitative information through individual interviews can reduce the impact of peers on respondents delivering expected and stereotypical answers [76].

However, as study participants were not blinded to the research question, there is a possibility that survivors who completed the interview had a strong positive interest in nutrition or had more profound nutrition-related complications. As participants were required to express their interest in taking part in the present study via email or phone, this may have unintentionally excluded cancer survivors with limited digital literacy or hearing impairments. Additionally, this study exclusively recruited esophageal and gastric cancer survivors who received treatment and resided in the Republic of Ireland, highlighting the current unmet nutrition care needs within the Irish health system (Health Service Executive). Thus, the transferability of these findings to other countries may be limited due to variations in healthcare systems, nutritional support services, and cancer care pathways. Also, we acknowledge that the ethnic homogeneity of our sample may limit the generalizability of the findings, as cultural differences can influence caregiving roles and burden, self-care practices, and coping strategies among cancer survivors [77,78]. The low response rate to the recruitment via advertising the study in the UGI clinic newsletter may have been influenced by several factors. First, the study advertisement was distributed alongside a separate postal survey, which may have led to survey fatigue or confusion about the purpose of communication [78]. Additionally, the psychological burden associated with reflecting on difficult experiences and thinking about cancer may have impacted participation [79]. Although obtaining data on reasons for non-response and withdrawal could have provided valuable insights into the low uptake of the postal advertisement and early withdrawals, ethical considerations regarding this must be acknowledged. As participants were informed about the voluntary nature of the study and their right to withdraw at any time without providing a reason, it was neither appropriate nor ethically permissible to follow up with individuals who chose not to participate in order to inquire about their reasons for non-response. Another limitation is the possibility of recall bias in long-term survivors, which may impact the findings of the present study [65]. Finally, as caregivers’ perspectives were included in an exploratory manner, the thematic analysis reflects the experiences most relevant to the survivors and may not fully capture the broader range of caregivers’ unmet needs.

## 5. Conclusions

The findings of the present study highlighted the need for more dietitian support pre-operatively, during the acute recovery phase, and extending into long-term survivorship. Additionally, the present study highlighted the importance of offering timely, personalized, and practical dietary information to survivors and their caregivers. Furthermore, the findings highlight the need for peer support and additional resources to help survivors and their caregivers effectively manage the long-term effects of cancer. This information can guide the development of tailored nutrition care programs for esophageal and gastric cancer survivors.

## Figures and Tables

**Figure 1 nutrients-17-01567-f001:**
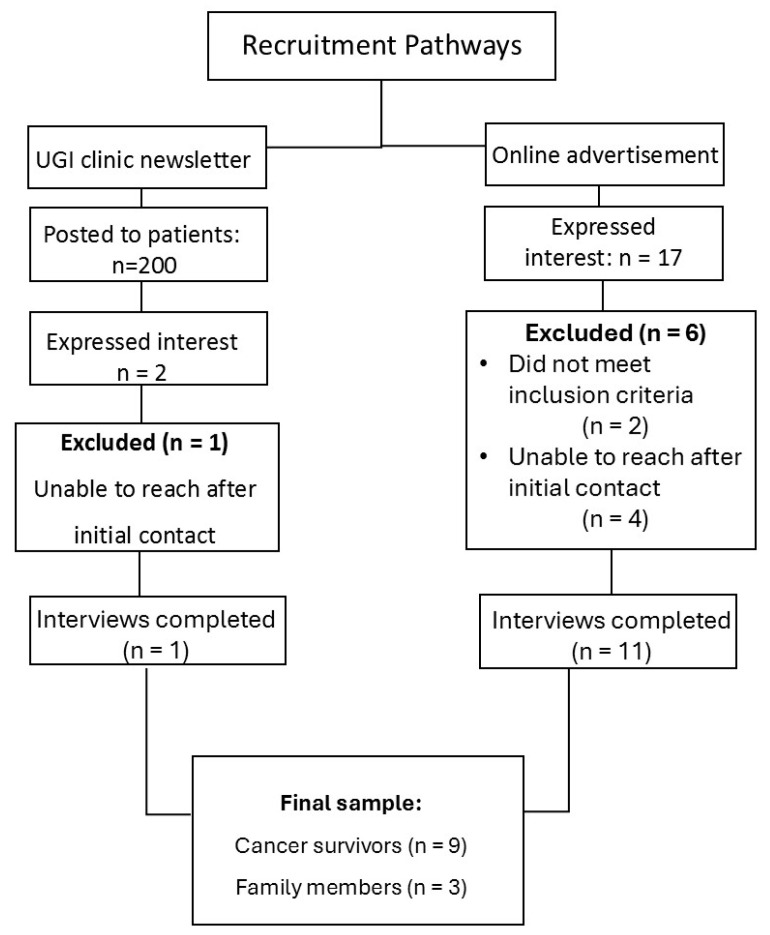
Recruitment flow diagram.

**Figure 2 nutrients-17-01567-f002:**
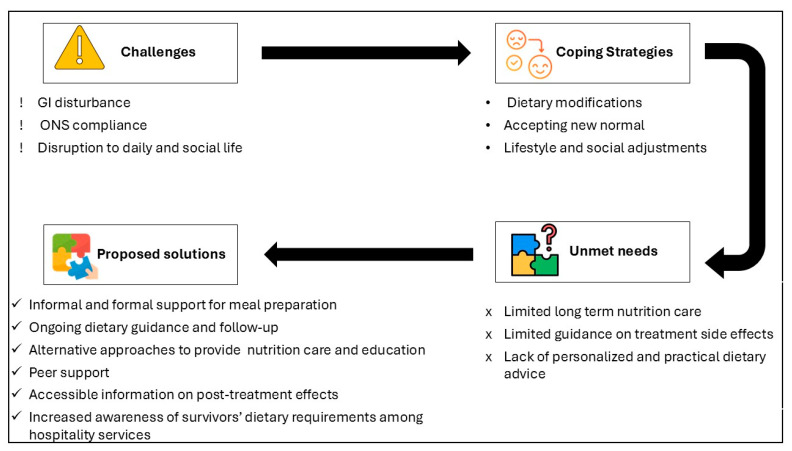
Conceptual diagram of challenges, unmet needs, and proposed solutions.

**Table 1 nutrients-17-01567-t001:** Characteristics of cancer survivors and family members participating in interviews.

Participant Code	Age	Gender	Cancer Diagnosis	Time Since Surgery (Months)
P1	76	M	Esophageal	108
P2	56	M	Esophageal	24
P3	63	M	Esophageal	24
P4	34	M	Esophageal	23
P5	51	F	Gastric	24
P6	71	M	Esophageal	62
P7 ^1^	60	F	Esophageal	62
P8	62	F	Gastric	15
P9	58	M	Esophageal	112
P10 ^1^	50	F	Esophageal	NA ^2^
P11	70	M	Esophageal/Gastric	87/20
P12 ^1^	55	M	Esophageal	NA ^2^

^1^ Interviewed family member, age, and gender represent the interviewee’s age and gender. ^2^ The patient did not undergo surgical resection.

**Table 2 nutrients-17-01567-t002:** Themes and subthemes.

Theme	Subthemes	Codes
Nutrition-related challenges and complications	Gastrointestinal related complications	Gastrointestinal symptoms
		Barriers to ONS compliance
	Impact on individuals’ daily life	Meal preparation and altered meal plans
		Limited food choices
		Disturbed daily and social life
Experiences with dietetic service	Access and effectiveness	Accessibility of dietetic service
		Effectiveness of dietitian visits
		Continuation and frequency of dietetic follow-up
	Information needs	Tailored and practical dietary advice
		What to expect post-treatment
		Additional educational material and methods
		Reiterating dietary advice
Coping strategies	Practical strategies	Dietary modifications
		Accepting new normal
		Support for meal preparation
		Consideration for eating out
	Peer and external support	Peer support
		Support for carers

ONS, oral nutritional supplement.

## Data Availability

The original contributions presented in this study are included in the article. Further inquiries can be directed to the corresponding author.

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
