# Peer review of "“One Size Doesn’t Fit All”: Nutrition Care Needs in Esophageal and Gastric Cancer Survivors—A Qualitative Study"

_nutrients, 2025, doi:10.3390/nu17091567_

Round 1

Reviewer 1 Report

Comments and Suggestions for Authors

Dear Authors,

Congratulations for writing the manuscript. Nevertheless, I have a few suggestions to improve the article:

  • In the introduction section, upper GI cancers are described, while included in your article only adults with esophageal and stomach cancers were included. Please specify this in the abstract and title of the article as well, as the nutritional needs of patients with esophageal cancer are significantly different from those arising in patients with pancreatic cancer for example.
  • I would suggest adding a section about the pathophysiological mechanisms behind the nutritional deficiencies in the studied cancers.
  • The Results section is too long and the readers can easily miss out important points, I suggest reorganizing and restructuring this section.

Thank you for the opportunity to review this manuscript and I am looking forward to receiving the modified version of the manuscript.

Author Response

Dear reviewer,

Thank you very much for taking the time to review our manuscript. Please find our detailed responses attached, along with the corresponding revisions highlighted in track changes in the revised manuscript.

Reviewer 2 Report

Comments and Suggestions for Authors

The article presented to us by our research colleagues has the aim of taking stock of the difficulties of patients interviewed by telephone by a dietician researcher. They were all united by the intervention, conducted on the upper G.I. tract of the digestive system. Regarding the paper, in our opinion, it should be a bit thinned out by trying to concentrate those that were the statements extracted from the telephone calls. Otherwise the paper is a bit heavy to read. It is also advisable to concentrate certain interventions in third level centers where in the oncology team there is certainly the figure of the nutritionist (doi.org/10.3390/nu17010188 to be read and cited in the bibliography) who obviously must take charge of the patient already at the time of diagnosis and then will have to follow him with follow-ups in the post-operative period. We know perfectly well that patients treated with neoadjuvant, adjuvant, surgical therapies or only systemic chemotherapies present a significant alteration of tastes that will affect their food intake, as it is equally true that the reconstruction of the digestive system will bring with it other problems related to the capacity of containment, therefore the nutritionist will have to take charge of the problems to find the best solution for each patient. It is impossible to tar everyone with the same brush, Excellent English, good bibliography, good bibliography

Author Response

Dear Reviewer,

Thank you very much for taking the time to review our manuscript. Please find our detailed responses attached, along with the corresponding revisions highlighted in track changes in the revised manuscript.

Reviewer 3 Report

Comments and Suggestions for Authors

This study investigates the nutritional care needs of Irish survivors of upper GI cancer and their caregivers. The authors conducted interviews with 12 participants, identifying three main themes: nutritional challenges, experiences with dietetic services, and coping strategies. I applaud the authors on this well written manuscript on such an important topic. While the study is clinically relevant and addresses a geographic gap in survivorship research, I have the following concerns and suggestions for improvement:

  • Including a recruitment flow diagram would be beneficial, showing the number of individuals who saw the advertisement, were screened, deemed eligible, interviewed, and reasons for non participation.
  • In the manuscript, its reported that data collection continued until saturation was reached; however there is no clear definition of what that means. 
  • Given that participants were mainly recruited through social media ads and cancer charity platforms, which may exclude older, digitally disconnected survivors or those not affiliated with advocacy groups, who are likely to have unmet dietetic needs. In my opinion, this is a significant limitation that is not acknowledged.
  • Since all interviews were conducted by a practicing dietitian, participants might have been hesitant to openly critique dietetic services. The study would benefit from reflexive notes detailing how the interviewer addressed role related bias (interviewer bias in this case).
  • Only three caregivers were interviewed, yet their quotes are included in thematic conclusions. It would be better to either highlight them in a different section or clearly present it as exploratory.
  • I recommend adding a conceptual diagram that links challenges, unmet needs, and proposed solutions; a simple timeline graphic would also enhance readability.
  • It is important to acknowledge the lack of ethnic diversity and the potential exclusion of participants without internet access or those with hearing impairments.

Author Response

(The authors gave the same response as above.)

Round 2

Reviewer 2 Report

Comments and Suggestions for Authors

The paper that we have the honor of reading for the second review is much more understandable than the first version of which the original trace was lost. The work carried out in your department could be defined as commendable. We understand that there are increasingly greater shortages of personnel in the health sector which obviously have repercussions on patients especially if fragile such as cancer patients. In light of this, however, you have completed excellent research that it is important to promote to a wider audience of colleagues who will be able to be informed on how to deal with important problems such as nutritional ones that still not many think about endorsement publication